# Genome-Wide Association Study on the Content of Nucleotide-Related Compounds in Korean Native Chicken Breast Meat

**DOI:** 10.3390/ani13182966

**Published:** 2023-09-20

**Authors:** Minjun Kim, Jean Pierre Munyaneza, Eunjin Cho, Aera Jang, Cheorun Jo, Ki-Chang Nam, Hyo Jun Choo, Jun Heon Lee

**Affiliations:** 1Division of Animal and Dairy Science, Chungnam National University, Daejeon 34134, Republic of Korea; kimmin6023@gmail.com (M.K.); inezajp@gmail.com (J.P.M.); 2Department of Bio-AI Convergence, Chungnam National University, Daejeon 34134, Republic of Korea; eunjjinie@gmail.com; 3Department of Applied Animal Science, College of Animal Life Science, Kangwon National University, Chuncheon 24341, Republic of Korea; ajang@kangwon.ac.kr; 4Department of Agricultural Biotechnology, Center for Food and Bioconvergence, Research Institute of Agriculture and Life Sciences, Seoul National University, Seoul 08826, Republic of Korea; cheorun@snu.ac.kr; 5Department of Animal Science and Technology, Sunchon National University, Suncheon 57922, Republic of Korea; kichang@scnu.ac.kr; 6Poultry Research Institute, National Institute of Animal Science, Rural Development Administration, Pyeongchang 25342, Republic of Korea

**Keywords:** Korean native chicken, meat flavor, nucleotide-related compounds, genome-wide association study, candidate gene

## Abstract

**Simple Summary:**

Recently, the consumption of high-quality meat has increased. Meat flavor is the factor that most significantly contributes to meat quality, making meat flavor an important performance trait in the chicken industry. Although chicken breeding has achieved fast-growing and high-quantity meat production so far, it is still necessary to improve the meat flavor of chicken through selection. Nucleotide-related compounds generate tastes such as umami and primarily determine meat flavor. In this study, for the first time, we identified candidate genes regulating nucleotide-related compounds included in chicken breast meat using a genome-wide association study. These findings will promote the selection of chicken to produce more tasty meat.

**Abstract:**

Meat flavor is an important factor that influences the palatability of chicken meat. Inosine 5′-monophosphate (IMP), inosine, and hypoxanthine are nucleic acids that serve as taste-active compounds, mainly enhancing flavor in muscle tissue. For this study, we performed a genome-wide association study (GWAS) using a mixed linear model to identify single-nucleotide polymorphisms (SNPs) that are significantly associated with changes in the contents of the nucleotide-related compounds of breast meat in the Korean native chicken (KNC) population. The genomic region on chicken chromosome 5 containing an SNP (rs316338889) was significantly (*p* < 0.05) associated with all three traits. The trait-related candidate genes located in this significant genomic region were investigated through performing a functional enrichment analysis and protein–protein interaction (PPI) database search. We found six candidate genes related to the function that possibly affected the content of nucleotide-related compounds in the muscle, namely, the *TNNT3* and *TNNT2* genes that regulate muscle contractions; the *INS*, *IGF2*, and *DUSP8* genes associated with insulin sensitivity; and the *C5NT1AL* gene that is presumably related to the nucleotide metabolism process. This study is the first of its kind to find candidate genes associated with the content of all three types of nucleotide-related compounds in chicken meat using GWAS. The candidate genes identified in this study can be used for genomic selection to breed better-quality chickens in the future.

## 1. Introduction

Meat consumption has increased worldwide over the past century [1] largely because of economic growth, urbanization, and developments in meat production technology [2]. On the other hand, in recent years, meat consumers have shown a preference for healthy and high-quality meat [3,4,5,6]. Meat consumers also prefer chicken meat because of its lower price, higher protein content, and low fat content compared to other meat [7,8,9]. Meat quality is a complex trait that is defined by factors such as pH, color, drip loss, tenderness, intramuscular fat content, water-holding capacity, abdominal and subcutaneous fat proportions, and flavor [10,11,12,13]. Multiple studies have shown that various traits positively correlate with meat quality, influencing consumer acceptance and meat price.

Taste and flavor, regarded as the most important traits of meat quality, are influenced by nucleotide-related compounds such as inosine 5′-monophosphate (IMP), which is the predominant nucleotide in meat [14]. IMP is also a representative taste-active precursor of meat [15]. It is degraded to inosine by 5′-nucleotidase; inosine is then converted into hypoxanthine through a nucleoside phosphorylase reaction. The nucleic acid components from the decomposition of IMP are also good indicators of meat palatability [16].

Similar to many other meat quality traits, meat flavor is regulated by numerous genes [17]; however, the improvement of these traits using the conventional selection method has been ineffective because of their low heritability [18]. Moreover, measurements of meat quality characteristics require the slaughtering of target animals, which is expensive and time consuming [19]. Genomic selection based on single-nucleotide polymorphisms (SNPs) in candidate genes could improve meat quality traits [20,21]. In livestock, a genome-wide association study (GWAS) is typically used to identify markers associated with phenotypes of interest, including weight, plumage color [22,23,24,25], disease [26], and other economically important traits. GWASs have also revealed candidate sites and genes for meat quality, which are key economic traits for livestock and poultry species.

Korean native chicken (KNC) is a purebred indigenous chicken population in Korea. Based on the plumage color, KNCs are categorized into five lines: white (KNC-W), black (KNC-B), yellow-brown (KNC-Y), gray-brown (KNC-G), and red-brown (KNC-R) [27,28,29,30]. The KNC breed is known for its characteristics regarding high meat quality. Many meat consumers prefer meat from KNC rather than meat from commercial broilers because of the unique taste and flavor of KNC meat [31,32,33]. The KNC-R line is particularly important, considering that it was selected by the National Institute of Animal Science (NIAS) in Korea for a crossbreeding combination to make Woorimatdag commercial chicken based on scientific evidence of its superior meat quality and productivity [10,34,35].

Previous studies on cattle, pigs, and chickens have used various methods to identify multiple genes related to IMP [16,17,36,37,38,39]. For example, *AMPD1* encodes adenosine monophosphate (AMP) deaminase, which is the enzyme that generates IMP from AMP in chicken [36]. Additionally, a group of regulatory genes, including *BMPR2* and *PTEN*, which are involved in IMP metabolism and muscle development, was identified in a gene expression study regarding chicken [17]. However, there remains a need for a GWAS to identify key genes that have a strong impact on IMP and its degradation product content in chicken muscle. Although meat flavor active compounds are important traits in livestock, the chicken quantitative trait locus database (chicken QTLdb) does not contain association information regarding the contents of nucleotide-related compounds in chicken muscle [40]. For this study, in order to identify the candidate genomic regions and putative candidate genes affecting meat quality based on the contents of nucleotide-related compounds in breast muscle, 637 birds from the KNC-R population were included in a GWAS. This study’s results will be useful for chicken breeders, enabling them to better understand underlying biological processes and identify the potential genes and markers that could be used in breeding programs to improve meat quality in chickens.

## 2. Materials and Methods

### 2.1. Animals

In this study, we used KNC-R line chickens maintained as a pure line at Poultry Research Institute in NIAS (National Institute of Animal Science) in Korea. The experimental chickens were selected from each generation out of four generations (1 generation is 52 weeks) to make up 637 birds in total. The number of birds and genders in each generation used for this experiment are described in Appendix A. For population maintenance, 210 females and 30 males were selected as the parents for the development of a new generation every year. The selection index for selecting birds for the next generation did not include the nucleotide-related compounds, meaning that selection did not affect the results of our analysis, which was carried out using data generated in this population.

All chickens of each generation were hatched on the same day. The chickens were raised in battery cages (1.16 m × 0.58 m, 0.67 m^2^/10 heads) in the windowless chicken house of the Poultry Research Institute from hatch to 10 weeks of age. In the growth stage, the chickens were fed a starter diet (ME: 2850, CP: 19.0) from hatch up until 6 weeks of age and a grower diet (ME: 2820, CP: 14.5) from 7 to 10 weeks of age. The feed and water were provided ad libitum. Feed and the environmental conditions (temperature, humidity, housing conditions and stocking density, etc.) applied to the chickens were maintained consistently.

At 10 weeks of age, the experimental birds were fainted via CO_2_ treatment after blood sampling. After exsanguination by the neck cut method, carcass weight and breast meat were collected from the eviscerated carcasses. The separated breast meat was stored at −80 °C for analysis. The collected blood samples were stored at −20 °C until DNA extraction. Our animal care and experimental protocols followed the relevant guidelines formulated by the Institution of Animal Care and Use Committee of the NIAS (Approval number: NIAS 20212219) and Animal Ethics Committee of Chungnam National University (Approval number: 202209A-CNU-141).

### 2.2. Phenotype Measurement and Preprocessing

For phenotype data collection, we used nuclear magnetic resonance spectroscopy to measure the IMP, inosine, and hypoxanthine contents in the breast meat samples [41]. The measured values were calculated in units of mM. Then, phenotypic values were normalized using a log, cube, or square root scaling method. The normalizing method that produced the highest Shapiro–Wilk *p*-value for normalized data among the methods was chosen. For outlier trimming, values that were larger or smaller than three standard deviations from the mean value of the data were removed. Finally, 635 IMP and inosine contents and 637 hypoxanthine contents were filtered and used in an association test. R software v 3.6.3 was used for preprocessing.

### 2.3. Genotyping and Preprocessing

A commercial kit (GeNet Bio, Daejeon, Korea) was used to extract DNA from the blood samples. Using an Illumina chicken 60 K SNP chip (Illumina Inc., San Diego, CA, USA), genotype data containing 57,636 SNPs were generated from the extracted DNA samples. Using PLINK1.9 v 1.90b5.2 software [42], SNPs with a call rate <0.9, minor allele frequency <0.01, and Hardy–Weinberg test *p*-value < 10^−6^ were filtered out prior to analysis. Finally, 44,842 variants remained and were used for the association test.

### 2.4. Construction of Genetic Relationship Metrics and GWAS

Before conducting the GWAS, we constructed the genetic relationship metrics of 637 birds from the KNC-R population using a genome-wide complex trait analysis (GCTA) [43] tool kit v 1.93, genetic relationship metrics, and the preprocessed phenotype. We calculated the heritability of each trait through REML analysis using GCTA software. The GWAS was conducted using a mixed linear model of the GCTA based on the leave-one-chromosome-out method. The following model was used:$$y=a+bx+g^{-}+e$$
where y is the adjusted phenotypic value using covariates such as the sex, generation group, and carcass weight of each tested animal, as well as the PC1 and PC2 of a principal component analysis according to genotypes; a is the mean of the phenotypic records; b is the additive effect of the tested SNP marker; x is the genotype of the SNP; $g^{-}$ is the effect of all SNPs that were calculated using the genetic relationship metrics, except for SNPs on the chromosome undergoing analysis; and e is the residual effect vector. After the analysis, the tested SNPs were assessed using a 5% significance threshold of the Bonferroni-corrected *p*-value to identify significantly associated SNPs. To visualize the results, the trend of marker significance was summarized in a Manhattan plot. We defined a significantly associated genomic region of 1 Mb centered around the position of the most significant SNP and then used that region for functional annotation analysis.

### 2.5. Gene Annotation and Functional Annotation Analysis

The genes located in the significant genomic region were retrieved from GRCg6a 106 version in the Ensembl genome database using BioMart [44]. These genes were used for functional annotation analysis. The annotation of gene ontology (GO) terms and Kyoto Encyclopedia of Genes and Genomes (KEGG) pathways was conducted for genes in the significant region using the ClusterProfiler [45] package in R. Only significant (*p* < 0.05) GO terms and KEGG pathways were reported. All annotation procedures were performed using the *Gallus gallus* database. Additional literature investigations for each gene function were also conducted. To investigate gene function, the interactions of proteins encoded by each gene with other proteins were analyzed using the STRING v 11.5 (http://string-db.org/; accessed on 1 April 2023) database with default settings.

## 3. Results

### 3.1. Basic Statistics of Phenotypic Values

Table 1 shows the descriptive statistics of the contents of nucleotide-related compounds measured in the breast meat of the KNC-R population. The mean and standard deviation of the subgroups’ phenotypic values are shown in Appendix A, and the measured phenotypic values are summarized via a histogram and quantile–quantile plot in Figure 1. Compared with the mean values, the phenotypic molecule content was highest for IMP at approximately 4.97 mM, followed by inosine and hypoxanthine at approximately 1.2 and 0.44 mM, respectively. Although the amount of data was sufficiently large for the assessment of all traits, the data distribution was not normally distributed and displayed skewness.

Therefore, the phenotypic values of IMP, inosine, and hypoxanthine were transformed using log, square root, and cube scaling, respectively. After transformation, inosine values exhibited a normal distribution (*p* > 0.05), but other traits did not exhibit a normal distribution according to the Shapiro–Wilk test. Inosine had a heritability of >0.5; IMP and hypoxanthine had heritability values of approximately 0.25 and 0.24, respectively.

### 3.2. GWAS Results and the Candidate Genomic Region

Based on the Bonferroni-corrected genome-wide significance threshold, 46 SNPs were significant (adjusted *p* < 0.05) for the inosine trait. All significant SNPs were located on chromosome 5 (10.7–18.8 Mb). Among the significant SNPs, 23 markers had a *p*-value < 10^−5^. The most significant SNP was rs316338889 (*p* = 5.83 × 10^−27^), which was located at 14,370,082 bp on chromosome 5. With respect to IMP, 17 SNPs were significant (adjusted *p* < 0.05). Two SNPs had a *p*-value < 10^−5^, including rs316338889, which was the most significant SNP (*p* = 1.62 × 10^−16^). Sixteen markers of seventeen significant SNPs in IMP were also common among the significant SNPs in inosine. In the GWAS results regarding hypoxanthine content, only one SNP (rs316338889) had a higher significance (adjusted *p* < 0.05) than the threshold; its *p*-value was 1.53 × 10^−9^. Notably, the rs316338889 SNP was significant in all three traits as the lead SNP, but none of the other genomic regions showed significance in the GWAS results. Manhattan plots and detailed GWAS results for each SNP are shown in Figure 2 and Appendix A, respectively. Consequently, we defined a significant genomic region that was associated with all three traits from 13,870,082 bp to 14,870,082 bp on chromosome 5; this comprised a 1 Mb region surrounding the lead SNP. Based on the GRCg6a 106 version genome annotation, the area extended from the *ASCL2* gene region to the *TOLLIP* gene region.

### 3.3. Gene Annotation and Functional Annotation Analyses

The following 16 protein-coding genes were included in the significant genomic regions: *ASCL2*, *TH*, *INS*, *IGF2*, *MRPL23*, *TNNT3*, *LSP1P1*, *TNNI2*, *SYT8*, *C5NT1AL*, *CTSD*, *IFITM10*, *DUSP8*, *MOB2*, *BRSK1*, and *TOLLIP*. We used these genes to perform GO analysis, which revealed 2 significant (*p* < 0.05) GO terms in the molecular function category, 11 terms in the biological process category, and 12 terms in the cellular component category. Only GO terms related to the target traits are listed in Table 2. Among the results, there were four GO terms related to hormone activity, which was enriched in the *INS* and *IGF2* genes. There were 11 GO terms related to the muscle fiber and contraction process, which was enriched in the *TNNT3* and *TNNI2* genes. None of the KEGG pathways were significantly enriched in genes in the significant genomic region. Analysis of genes forming a network with genes in the protein–protein interaction (PPI) database revealed a gene network containing *C5NT1AL* (also known as *LOC770392*); we inferred that this gene was related to the contents of nucleotide-related compounds. This network consisted of 11 proteins (PNP, GMPR, NUDT12, UPP1, ADA, DTYMK, CMPK2, ITPA, ATIC, DUT, and C5NT1AL) that were functionally clustered with high confidence and an interaction score >0.9. The groups of proteins the featured C5NT1AL in the network were enriched in the metabolic (gga01100), purine metabolism (gga00230), and pyrimidine metabolism (gga00240) pathways in the KEGG database. Additionally, groups of proteins were enriched in the GO terms of adenosine metabolic process (GO:0046085), nucleoside phosphate catabolic process (GO:1901292), and nucleotide metabolic process (GO:0009117), which were included in the biological process category of the GO database (Appendix A).

## 4. Discussion

### 4.1. GWAS and the Relationships among IMP, Inosine, and Hypoxanthine in Chicken Muscle

In this study, we aimed to identify the genetic factors and markers related to the contents of nucleotide-related compounds, which are flavor precursors in the breast meat of KNC. In vertebrate muscle, IMP, inosine, and hypoxanthine act as precursors of each other during synthesis via nucleotide metabolism [16,46]. Consistent with these relationships, the GWAS results for each of the three traits revealed the same significant genome region on chromosome 5; most of the significant SNP markers were common to all three traits. These results are in agreement with the findings of a previous association study [16] regarding IMP content in bovine longissimus thoracic muscle and its degradation products. The previous association study [16] identified that the effect of a single gene encoding a particular nucleotidase in bovine muscle may have influenced the IMP, inosine, and hypoxanthine contents, leading to the observed GWAS results. We assumed that the association trend in our study was also derived from genetic factors that simultaneously affected the content of the three nucleotide-related compound molecules. There was a possibility that single or multiple causative genes existed in the genomic region near rs316338889, the common significant SNP in all three traits. Accordingly, the neighboring variants in a linkage disequilibrium were significantly clustered together in the association test. Therefore, we assumed that a single genomic region was significantly associated with all three traits; we subsequently conducted a gene function analysis to identify candidate genes. Furthermore, the difference in SNP significance level among the three traits was presumably related to the different distribution trends of phenotypes used for the statistical analysis, which affected the performance of the statistical test used to estimate SNP effects [47]. In particular, as shown in Figure 1, the estimation efficiency of the statistical model was appropriate for the evaluation of the SNP effect on inosine because the distribution of inosine content was less skewed than the distributions of the other two traits. This may explain the higher significance of inosine-related SNPs.

### 4.2. Functional Annotation and Candidate Genes—TNNT3 and TNNI2

Based on the GWAS results, we defined a 1-Mb significant genomic region on chicken chromosome 5 that was associated with the contents of nucleotide-related compounds, then obtained a list of 16 protein-coding genes located in that region. GO analysis involving these genes yielded 25 GO terms that were enriched in five genes (Appendix A). Among these five candidate genes, *TNNT3* and *TNNI2* were annotated with GO terms related to muscle contraction and muscle fiber (Table 2). *TNNT3* and *TNNI2* genes are expressed in the fast skeletal muscle fiber type in vertebrates; they encode troponin T and troponin I, respectively [48,49,50]. Troponin T and troponin I are subunits of the troponin complex in muscle fiber; troponin plays a major role in muscle contraction by regulating the myosin-actin interaction in striated muscles [51]. Previous studies [51,52,53] have shown that troponin regulates myosin ATPase activity. Because myosin ATPase is the main enzyme that dephosphorylates adenosine triphosphate (ATP) to adenosine diphosphate (ADP) in muscles, the activity of this enzyme directly affects the content of ATP degradation products in muscle tissue. Additionally, ADP produces AMP with assistance from nucleotidase, and AMP subsequently generates IMP via the AMP deamination reaction; therefore, the IMP content can change in muscle fiber [54]. In chickens, ATP and IMP contents are both regulated by the degradation of ATP to IMP and the process of synthesizing ATP from IMP (i.e., the purine nucleotide cycle) [14]. Therefore, the *TNNT3* and *TNNI2* genes, which can affect myosin ATPase activity in fast skeletal muscle fibers (e.g., breast muscle in chickens), may have contributed to the differences observed in this study regarding the contents of IMP and its degradation products.

### 4.3. Functional Annotation and Candidate Genes—INS, IGF2, and DUSP8

The GO analysis results included terms related to hormone activity that were enriched in the *INS* and *IGF2* genes. The *INS* gene, encoding insulin, is reportedly related to muscle fiber density and growth traits in chickens [55,56]. Insulin is a representative enzyme that regulates glucose disposal, storage, and oxidation in skeletal muscle [57]. The *IGF2* gene, encoding an insulin-like growth factor, reportedly induces glucose uptake in vertebrates and plays a major role in glucose metabolism [58,59]. In the present study, these two genes were significantly annotated in terms of hormone activity and glucose metabolic processes in the GO analysis results. Additionally, the *DUSP8* gene, which possessed the most significant SNP in this study, participates in the regulation of glucose tolerance and insulin sensitivity [60]. Importantly, the regulation of insulin sensitivity and glucose metabolism leads to changes in the amount of ATP synthesized from glucose via glycolysis in muscle; this ATP is degraded to IMP after energy production in chicken muscle [14,61,62,63]. Therefore, the functions of the *INS*, *IGF2*, and *DUSP8* genes likely have major effects on the content of nucleotide-related compounds in the muscle of KNC-R chickens.

### 4.4. Functional Annotation and Candidate Genes—C5NT1AL

The *C5NT1AL* gene in the significant genomic region encodes a cytosolic 5′-nucleotidase 1A-like enzyme that remains uncharacterized. According to the KEGG pathway database, the *C5NT1AL* gene is involved in a nucleotide metabolism pathway (gga01232) that acts on nucleoside monophosphate; this pathway contributes to the degradation of IMP to inosine. Specifically, along with *NT5E* and *NT5C1A*, the *C5NT1AL* gene is registered in the KEGG enzyme entry (EC no. 3.1.3.5) that describes the reaction by which IMP is dephosphorylated into inosine. Thus, C5NT1AL may be a 5′-nucleotidase that participates in the regulation of IMP metabolism [16,17]. Additionally, PPI enrichment analysis revealed that C5NT1AL participates in purine metabolism through interactions with the GMPR, ITPA, PNP, and ADA proteins (Figure 3). GMPR and ITPA are enzymes involved in IMP production through the deamination of guanosine monophosphate and the dephosphorylation of inosine triphosphate, respectively [64,65]. In contrast, ADA is an enzyme that converts adenosine to inosine, thereby increasing the amount of inosine; PNP is a 5′-nucleotidase or phosphorylase that degrades the inosine generated by ADA into hypoxanthine [66,67]. Therefore, C5NT1AL is presumably a protein that significantly interacts with enzymes contributing to this cascade of purine metabolism in chickens. The results of our KEGG and GO analyses of the PPI database support the suspected *C5NT1AL* gene function and its effect on the traits (Appendix A). Therefore, the GWAS results and functional annotation analysis results in this study imply that the function of the *C5NT1AL*-encoded protein is related to IMP degradation, as well as the inosine synthesis and hypoxanthine synthesis processes. Overall, the *C5NT1AL* gene is a candidate gene that regulates changes in the contents of the nucleotide-related compounds of breast muscle in the KNC-R population.

This research identified positional candidate genes and conducted functional annotation for the identified genes. The significant variant identified through the GWAS was used to define the candidate genome region. However, this does not indicate that the change in the variant leads to the phenotypic differences because the SNP chip genotype was used in this study. Therefore, it is necessary to screen other variants within the identified candidate genes and validate them through further studies (e.g., by genotyping the variant to confirm its association with the phenotype). Moreover, integrated studies should be conducted to characterize the transcriptome and confirm the functions of the encoded proteins.

## 5. Conclusions

The flavor of chicken meat is influenced by IMP and its degradation products (inosine and hypoxanthine). We conducted a GWAS and detected a genomic region on chromosome 5 that had a significant effect on the contents of nucleotide-related compounds in breast meat. We identified the following candidate genes via functional annotation: *TNNT3*, *TNNI2*, *INS*, *IGF2*, *DUSP8*, and *C5NT1AL*. These genes showed a possible association with ATP regulation and nucleic acid catabolism. The results of this study will be useful for farmers breeding KNC, enabling them to produce high-quality meat in the future.

## Figures and Tables

**Figure 1 animals-13-02966-f001:**
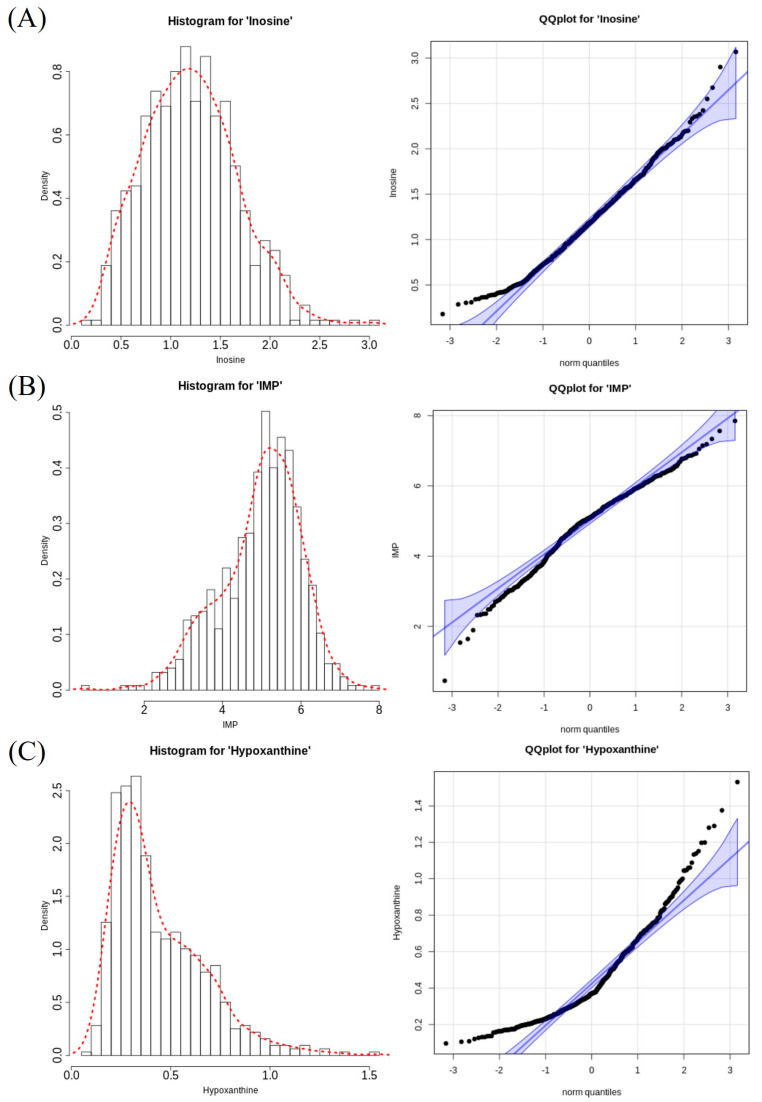
Phenotype distributions for the contents of nucleotide−related compounds measured in the chicken breast meat. (**A**–**C**) Histogram and quantile–quantile plots of the inosine, IMP, and hypoxanthine contents in chicken breast meat, respectively.

**Figure 2 animals-13-02966-f002:**
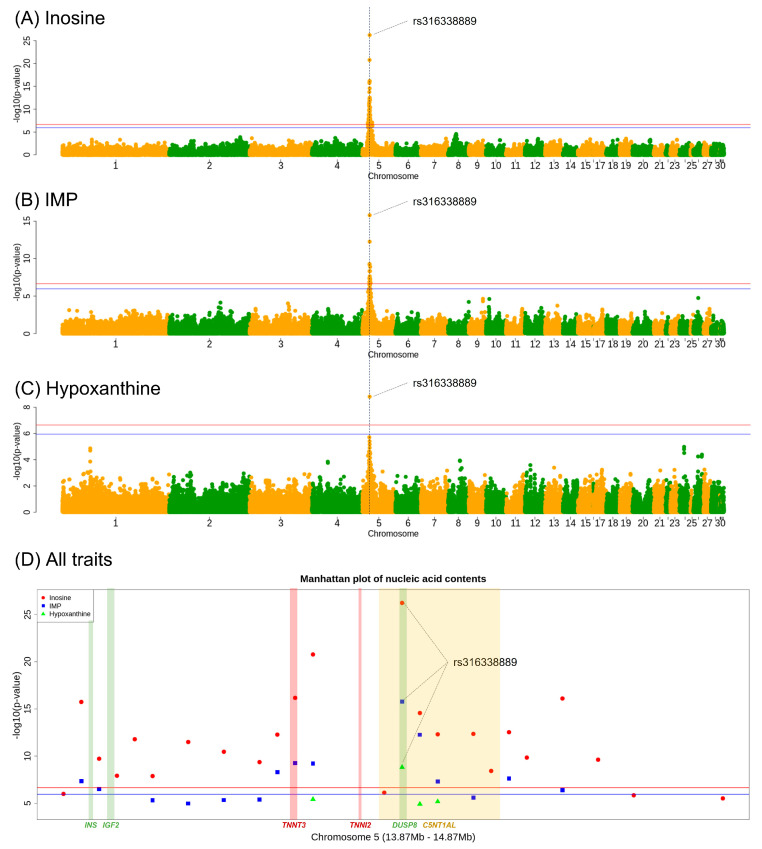
Genome-wide Manhattan plots of −log_10_(*p*-value) for significant SNPs. (**A**–**C**) show the association test results for inosine, IMP, and hypoxanthine contents, respectively. The *x*-axis indicates chromosome number, and the *y*-axis depicts −log_10_(*p*-value) to show the significance of the tested SNPs. The *y*-axis scale of (**A**–**C**) are different. The blue and red lines indicate the Bonferroni-corrected 5% and 1% significance threshold, respectively. The blue dotted line indicates the physical position of the most significant SNP of each trait; it shows that the same SNP (rs316338889) was the most significant SNP for all nucleotide-related compounds. (**D**) shows a regional plot of the loci associated with inosine, IMP, and hypoxanthine. The plotted region is the significant genomic region, which was defined as the 1-Mb region surrounding the most significant SNP identified in the GWAS. The points with different colors and shapes indicate the different significances of each SNP for each trait. The approximate positions of the *INS*, *IGF2*, *TNNT3*, *TNNI2*, *DUSP8*, and *C5NT1AL* genes are shown; points are colored to indicate the groups of genes with similar functional effects on each trait.

**Figure 3 animals-13-02966-f003:**
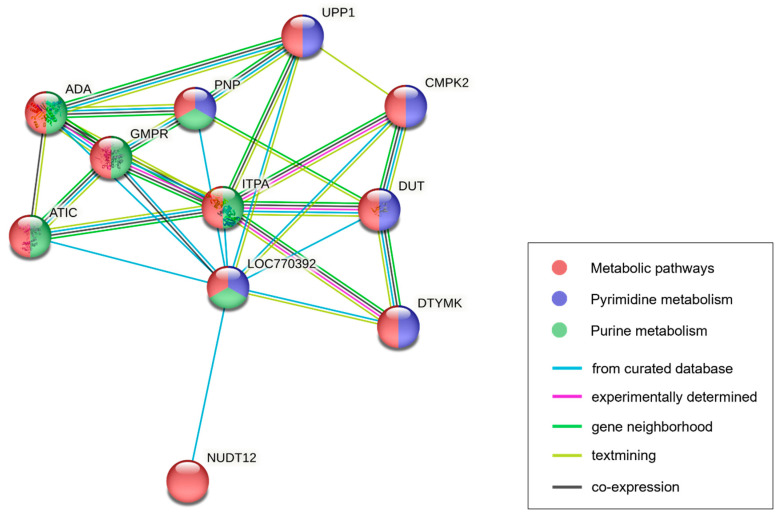
Protein–protein interactions of the *C5NT1AL* (*LOC770392*) gene in chicken. Each node and corresponding annotation indicate the protein (gene) involved in the network. Node colors indicate different KEGG pathways in which each protein is enriched. Edges between nodes indicate identified protein interactions. Edge color represents the source of the interaction in the database. Ten proteins had interactions with C5NT1AL; the corresponding genes were involved in metabolic pathways together with C5NT1AL.

**Table 1 animals-13-02966-t001:** Descriptive statistics and heritability estimates of nucleotide-related compounds measured in the chicken breast meat (unit: mM).

Traits	Mean	SD	CV (%)	Minimum	Maximum	Heritability	Trans
Inosine	1.2002	0.4647	0.3872	0.1780	3.0676	0.5197	Sqrt
IMP	4.9658	1.0253	0.2065	0.4489	7.8536	0.2577	Log
Hypoxanthine	0.4384	0.2266	0.5169	0.0960	1.5322	0.2493	Cube

SD, Standard deviation; CV, Coefficient of variation; Trans, Transformation methods; Sqrt, Square root scaling; Cube, Cube scaling; Log, Log scaling.

**Table 2 animals-13-02966-t002:** GO terms related to genes in the significant genomic region.

Category	ID	GO Term	*p*-Value	Gene Names
Molecular function	GO:0005179	hormone activity	0.0039	*INS*, *IGF2*
Biological process	GO:0006941	striated muscle contraction	0.0007	*TNNT3*, *TNNI2*
	GO:0006006	glucose metabolic process	0.0023	*INS*, *IGF2*
	GO:0019318	hexose metabolic process	0.0024	*INS*, *IGF2*
	GO:0005996	monosaccharide metabolic process	0.003	*INS*, *IGF2*
	GO:0006936	muscle contraction	0.0034	*TNNT3*, *TNNI2*
	GO:0003012	muscle system process	0.0046	*TNNT3*, *TNNI2*
Cellular component	GO:0005865	striated muscle thin filament	0.0005	*TNNT3*, *TNNI2*
	GO:0036379	myofilament	0.0005	*TNNT3*, *TNNI2*
	GO:0030017	sarcomere	0.0059	*TNNT3*, *TNNI2*
	GO:0044449	contractile fiber part	0.007	*TNNT3*, *TNNI2*
	GO:0030016	myofibril	0.0085	*TNNT3*, *TNNI2*
	GO:0043292	contractile fiber	0.0092	*TNNT3*, *TNNI2*
	GO:0015629	actin cytoskeleton	0.0189	*TNNT3*, *TNNI2*

## Data Availability

The data presented in this study are available upon request from the corresponding author.

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
