# Peer review of "Genome-Wide Association Study on the Content of Nucleotide-Related Compounds in Korean Native Chicken Breast Meat"

_animals, 2023, doi:10.3390/ani13182966_

Round 1

Reviewer 1 Report (Previous Reviewer 1)

none

Author Response

This is the authors' response to reviewer 2. We are sharing these comments and the responses to reviewer 1 as well.

< Response to reviewer 2 comments >

General comment.

KIM et al. collected phenotype data from 637 KNC-R line chickens of four generations which were normalized by a log, cube, or square root scaling method. This study utilized GCTA to identify genomic variants associated with IMP, inosine and hypoxanthine contents in breast meat of Korean native chickens. The study revealed a significant region on chromosome 5 containing a SNP (rs316338889) was associated with all three traits whereas INS, IGF2, and DUSP8 encode proteins that regulate insulin sensitivity and affect ATP content in muscle after gene annotation and functional analyses. This study may provide some insights into the genetic basis of meat quality in chicken.

Materials and methods

Comment 1. Line 101: ‘is discribed’ > ‘are discribed’

Thanks to the reviewer for checking detailed points in the manuscript. The sentence “is described” was revised to “are described” in line 101 of the revised manuscript.

Comment 2. Line 147-154: the mixed linear model didn’t show the adjusted phenotype or fixed effects, if sex and generation group were better as fixed effects not covariates.

We appreciate the reviewer’s comments on the explanation of the statistical model. We agree that the explanation of the “y” was insufficient. In the study, adjusted phenotype using the covariates were used for analysis. To make it clear, we modified the sentence “where y is the phenotypic value” in line 149 of the submitted manuscript to “where y is the adjusted phenotypic value using covariates which were the sex, generation group, carcass weight of each tested animal, PC1 and PC2 of a principal component analysis according to genotypes” in line 149-151 of the revised manuscript. And the sentence “To more accurately estimate the SNP effect, the sex, generation group, and carcass weight of each tested animal were used as covariates, along with PC1 and PC2 of a principal component analysis according to genotypes.” in lines 152-155 of the submitted manuscript was removed.

Results

Comment 3. Line 175-182: phenotype in Table1 and Figure1 is not clear whether before or after transformation.

Thanks for the reviewer’s comment. We agree that the manuscript can make readers confuse the measured data in the figure and table with the transformed data mentioned in the manuscript. We added “measured” in lines 175, 188, and 194 of the revised manuscript to specify the data in Figure 1 and Table 1 are measured phenotype. We put a line breaking in line 181 of the revised manuscript to separate the explanations of the measured data and the transformed data. Also, “(Table 1)” in line 186 of the submitted manuscript was removed. Basically, the sentence “Table 1 shows the descriptive statistics of the contents of nucleotide-related compounds measured in the breast meat of the KNC-R population.” in line 173 of the revised manuscript prevents readers from being confused.

Comment 4. Line 188: part of the minimum and maximum phenotypic values should be considered outliers (exceed mean ± 3 SD)

Thanks for the reviewer’s comment. Descriptive statistics in Table 1 show data before transformation. And the outlier trimming was conducted using the transformed data before applying the statistical model. Some phenotypic values were removed as outliers. The sentence “For outlier trimming, values that were larger or smaller than three standard deviations from the mean value of the data were removed. Finally, 635 IMP and inosine contents and 637 hypoxanthine contents were filtered and used in an association test” mentioned in line 128-130 of the submitted manuscript was the relative explanation.

Comment 5. Line 192-195: figure legend repeated as 'Histogram for hypoxanthine ' and 'hypoxanthine'.

Thanks for the comment on the figure. However, the Figure 2 in the submitted manuscript which can be downloaded from the current revision system does not have the issue which the reviewer pointed out.

Comment 6. Line 215: chromosome 28-33 are mixed in figure 2

Thanks for the detailed concern about the figure. We modified Figure 2 to solve the problem of showing chromosome number.

Comment 7. It better to shorten the part of gene functional annotation in discussion and increase sample collection, phenotype measurement and GWAS part.

Thanks for the suggestion of the reviewer about the content in the discussion part. We understand the importance of the research contents that the reviewer suggested. The results of GWAS are discussed in 4.1 in the manuscript. However, we think that the additional contents are not essential for the discussion part. Because this research was the association study conducted with the purpose of finding related genomic region and identifying candidate genes with traits. Therefore, we focused on the discussion of the GWAS, functional annotation results, and the limitations of the current study in this manuscript.

Comment 8. 4.2. Functional analysis,4.3 and 4.4,functional analysis may overstatement here, because the author did not carried out any function validation for candidate genes.

Functional analysis is not the same as “functional annotation analysis”

We appreciate the reviewer for correcting the important terminology. Since we conducted functional annotation analysis for the positional candidate genes, the expression “functional analysis” in this manuscript was changed to “functional annotation analysis” in the revised manuscript.

Additionally, the sentences in the 4.2, 4.3, and 4.4 of discussion section were written based on the results of functional annotation analysis and paper investigation. Since the research does not include functional validation for genes, the potential of the candidate genes was explained with references as a basis in the discussion part.

Comment 9. Discussion can also consider to include the limitation and further work of current study.

Thanks for the concern about the discussion point. We agree that the limitations of this study need to be clearly mentioned in the discussion part. Therefore, we modified the sentences “The use of candidate genes identified in this study for chicken selection requires further research to detect additional genetic variants with causal effects on the traits.” in line 349-350 of the submitted manuscript to “This research identified positional candidate genes and conducted functional annotation for the identified genes. The significant variant identified through the GWAS was used to define the candidate genome region. However, this does not indicate that the change in the variant leads to the phenotypic differences because the SNP chip genotype was used in this study. Therefore, it is necessary to screen other variants within the identified candidate genes and validate them through further studies, such as genotyping the variant to confirm its association with the phenotype.” in lines 350-356 of the revised manuscript. Currently, we have been conducting this further research work.

Comment 10. Conclusion should be shorten to highlight the most important points.

We agree with the idea of the reviewer on the conclusion. We modified the sentences “TNNT3 and TNNI2 encode protein complex subunits that can affect ATPase activity during skeletal muscle contraction, whereas INS, IGF2, and DUSP8 encode proteins that regulate insulin sensitivity and affect ATP content in muscle. Because ATP is converted to IMP via metabolism in muscles, regulation of ATP content is a key factor influencing changes in the contents of multiple nucleotide-related compounds. Finally, functional analysis showed that the C5NT1AL gene encodes an unknown protein related to nucleic acid catabolism” in the 358-364 of the submitted manuscript to “These genes showed possible association with ATP regulation and nucleic acid catabolism” in lines 364-365 of the revised manuscript.

Reviewer 2 Report (Previous Reviewer 2)

KIM et al. collected phenotype data from 637 KNC-R line chickens of four generations which were normalized by a log, cube, or square root scaling method. This study utilized GCTA to identify genomic variants associated with IMP, inosine and hypoxanthine contents in breast meat of Korean native chickens. The study revealed a significant region on chromosome 5 containing a SNP (rs316338889) was associated with all three traits whereas INS, IGF2, and DUSP8 encode proteins that regulate insulin sensitivity and affect ATP content in muscle after gene annotation and functional analyses. This study may provide some insights into the genetic basis of meat quality in chicken.

Materials and methods

1. Line 101: ‘is discribed’ > ‘are discribed’

2. Line 147-154: the mixed linear model didn’t show the adjusted phenotype or fixed effects, if sex and generation group were better as fixed effects not covariates.

Results

3. Line 175-182: phenotype in Table1 and Figure1 is not clear whether before or after transformation.

4. Line 188: part of the minimum and maximum phenotypic values should be considered outliers (exceed mean ± 3 SD)

5. Line 192-195: figure legend repeated as 'Histogram for hypoxanthine ' and 'hypoxanthine'.

6. Line 215: chromosome 28-33 are mixed in figure 2

7. It better to shorten the part of gene functional annotation in discussion and increase sample collection, phenotype measurement and GWAS part.

8. 4.2. Functional analysis, 4.3 and 4.4,functional analysis may overstatement here, because the author did not carried out any function validation for candidate genes.

Functional analysis is not the same as functional annotation analysis

9. Discussion can also consider to include the limitation and further work of current study.

10. Conclusion should be shorten to highlight the most important points.

The manuscript was well written, and may be benifit from mimor editing of English.

Author Response

< Response to reviewer 2 comments >

General comment.

KIM et al. collected phenotype data from 637 KNC-R line chickens of four generations which were normalized by a log, cube, or square root scaling method. This study utilized GCTA to identify genomic variants associated with IMP, inosine and hypoxanthine contents in breast meat of Korean native chickens. The study revealed a significant region on chromosome 5 containing a SNP (rs316338889) was associated with all three traits whereas INS, IGF2, and DUSP8 encode proteins that regulate insulin sensitivity and affect ATP content in muscle after gene annotation and functional analyses. This study may provide some insights into the genetic basis of meat quality in chicken.

Materials and methods

Comment 1. Line 101: ‘is discribed’ > ‘are discribed’

Thanks to the reviewer for checking detailed points in the manuscript. The sentence “is described” was revised to “are described” in line 101 of the revised manuscript.

Comment 2. Line 147-154: the mixed linear model didn’t show the adjusted phenotype or fixed effects, if sex and generation group were better as fixed effects not covariates.

We appreciate the reviewer’s comments on the explanation of the statistical model. We agree that the explanation of the “y” was insufficient. In the study, adjusted phenotype using the covariates were used for analysis. To make it clear, we modified the sentence “where y is the phenotypic value” in line 149 of the submitted manuscript to “where y is the adjusted phenotypic value using covariates which were the sex, generation group, carcass weight of each tested animal, PC1 and PC2 of a principal component analysis according to genotypes” in line 149-151 of the revised manuscript. And the sentence “To more accurately estimate the SNP effect, the sex, generation group, and carcass weight of each tested animal were used as covariates, along with PC1 and PC2 of a principal component analysis according to genotypes.” in lines 152-155 of the submitted manuscript was removed.

Results

Comment 3. Line 175-182: phenotype in Table1 and Figure1 is not clear whether before or after transformation.

Thanks for the reviewer’s comment. We agree that the manuscript can make readers confuse the measured data in the figure and table with the transformed data mentioned in the manuscript. We added “measured” in lines 175, 188, and 194 of the revised manuscript to specify the data in Figure 1 and Table 1 are measured phenotype. We put a line breaking in line 181 of the revised manuscript to separate the explanations of the measured data and the transformed data. Also, “(Table 1)” in line 186 of the submitted manuscript was removed. Basically, the sentence “Table 1 shows the descriptive statistics of the contents of nucleotide-related compounds measured in the breast meat of the KNC-R population.” in line 173 of the revised manuscript prevents readers from being confused.

Comment 4. Line 188: part of the minimum and maximum phenotypic values should be considered outliers (exceed mean ± 3 SD)

Thanks for the reviewer’s comment. Descriptive statistics in Table 1 show data before transformation. And the outlier trimming was conducted using the transformed data before applying the statistical model. Some phenotypic values were removed as outliers. The sentence “For outlier trimming, values that were larger or smaller than three standard deviations from the mean value of the data were removed. Finally, 635 IMP and inosine contents and 637 hypoxanthine contents were filtered and used in an association test” mentioned in line 128-130 of the submitted manuscript was the relative explanation.

Comment 5. Line 192-195: figure legend repeated as 'Histogram for hypoxanthine ' and 'hypoxanthine'.

Thanks for the comment on the figure. However, the Figure 2 in the submitted manuscript which can be downloaded from the current revision system does not have the issue which the reviewer pointed out.

Comment 6. Line 215: chromosome 28-33 are mixed in figure 2

Thanks for the detailed concern about the figure. We modified Figure 2 to solve the problem of showing chromosome number.

Comment 7. It better to shorten the part of gene functional annotation in discussion and increase sample collection, phenotype measurement and GWAS part.

Thanks for the suggestion of the reviewer about the content in the discussion part. We understand the importance of the research contents that the reviewer suggested. The results of GWAS are discussed in 4.1 in the manuscript. However, we think that the additional contents are not essential for the discussion part. Because this research was the association study conducted with the purpose of finding related genomic region and identifying candidate genes with traits. Therefore, we focused on the discussion of the GWAS, functional annotation results, and the limitations of the current study in this manuscript.

Comment 8. 4.2. Functional analysis,4.3 and 4.4,functional analysis may overstatement here, because the author did not carried out any function validation for candidate genes.

Functional analysis is not the same as “functional annotation analysis”

We appreciate the reviewer for correcting the important terminology. Since we conducted functional annotation analysis for the positional candidate genes, the expression “functional analysis” in this manuscript was changed to “functional annotation analysis” in the revised manuscript.

Additionally, the sentences in the 4.2, 4.3, and 4.4 of discussion section were written based on the results of functional annotation analysis and paper investigation. Since the research does not include functional validation for genes, the potential of the candidate genes was explained with references as a basis in the discussion part.

Comment 9. Discussion can also consider to include the limitation and further work of current study.

Thanks for the concern about the discussion point. We agree that the limitations of this study need to be clearly mentioned in the discussion part. Therefore, we modified the sentences “The use of candidate genes identified in this study for chicken selection requires further research to detect additional genetic variants with causal effects on the traits.” in line 349-350 of the submitted manuscript to “This research identified positional candidate genes and conducted functional annotation for the identified genes. The significant variant identified through the GWAS was used to define the candidate genome region. However, this does not indicate that the change in the variant leads to the phenotypic differences because the SNP chip genotype was used in this study. Therefore, it is necessary to screen other variants within the identified candidate genes and validate them through further studies, such as genotyping the variant to confirm its association with the phenotype.” in lines 350-356 of the revised manuscript. Currently, we have been conducting this further research work.

Comment 10. Conclusion should be shorten to highlight the most important points.

We agree with the idea of the reviewer on the conclusion. We modified the sentences “TNNT3 and TNNI2 encode protein complex subunits that can affect ATPase activity during skeletal muscle contraction, whereas INS, IGF2, and DUSP8 encode proteins that regulate insulin sensitivity and affect ATP content in muscle. Because ATP is converted to IMP via metabolism in muscles, regulation of ATP content is a key factor influencing changes in the contents of multiple nucleotide-related compounds. Finally, functional analysis showed that the C5NT1AL gene encodes an unknown protein related to nucleic acid catabolism” in the 358-364 of the submitted manuscript to “These genes showed possible association with ATP regulation and nucleic acid catabolism” in lines 364-365 of the revised manuscript.

This manuscript is a resubmission of an earlier submission. The following is a list of the peer review reports and author responses from that submission.

Round 1

Reviewer 1 Report

line 49: drop > drip

lines 54-55: please provide the number of animals of each line.

lines 62-63: does fat content and composition also influence meat taste? Please add a comment.

Line 88: and candidate genes > and putative candidate genes.

Line 98: Do you use all the 637 birds for GWAS? Filtering?

Lines 151-153: please delete.

Lines 165-166: how do you calculate heritability?

Figure 1: Please correct the headings of the histograms (A)-(B)-(C).

Line 177: why you don’t calculate also the Bonferroni adjusted p>0.01 and reported this threshold in Figure 2?

Figure 2: please write that the Y-axis of panels (A), (B), (C), are different.

Figure 2: Is rs316338889 located in an exon or in an intron (or in another noncoding region) of DUSP8 gene?

Line 191: Please indicate with more details the database you used because in the current release of NCBI (https://www.ncbi.nlm.nih.gov/genome/gdv/browser/genome/?id=GCF_016699485.2) and ENSEMBL (https://www.ensembl.org/Gallus_gallus/Gene/Summary?db=core;g=ENSGALG00010024637;r=5:13758675-13792985;t=ENSGALT00010060146) databases the gene DUSP8 and the marker rs316338889 (ENSEMBL only) are located outside the region you considered in figure 2(D).

Line 206: Have you also considered the noncoding genes in the significant genomic region?

Lines 230-233: please delete.

Line 248: rs31633888 > rs316338889

From line 260: have you considered all the protein-coding genes located within the significant genomic region for the functional analysis? Have you also considered the closest to the most significant markers?

none

Author Response

[ Response to the reviewer 1’s comments addressed by point-by-point ]

 * Please see the attached revised manuscript.

Comment 1. line 49: drop > drip

Response: Thanks for the reviewer’s correction. In line 49 of the revised manuscript, “drop loss” was modified to “drip loss”.

Comment 2. lines 54-55: please provide the number of animals of each line.

Response: Thank you for the reviewer’s suggestion about detailed information of KNC lines. Lines 54-55 of the manuscript described that the KNC breed consists of 5 lines before introducing the KNC-R line which was used in this study. Therefore, the number of maintained animals of each line is not essential to understand this research design. The number of animals (KNC-R line) used in this study was provided in the “Materials and Methods” section.

Comment 3. lines 62-63: does fat content and composition also influence meat taste? Please add a comment.

Response: Intramuscular fat content and fatty acid composition are the major factors influencing meat quality and taste. Therefore, in lines 48-52 of the manuscript, we explained them with reference. In addition, in lines 62-64 of the manuscript, we intentionally explained solely the nucleotide-related compounds affecting meat taste, because we used IMP, Inosine, and hypoxanthine for the phenotype in this study. Therefore, I carefully think that information about fat contents and composition is not included in the text.

Comment 4. Line 88: and candidate genes > and putative candidate genes.

Response: We agree with the suggestion of the reviewer. In line 88 of the revised manuscript, the word “candidate genes” was modified to “putative candidate genes”.

Comment 5. Line 98: Do you use all the 637 birds for GWAS? Filtering?

Response: Thank you for the comment. Genotype QC did not remove any sample. But in the case of IMP and Inosine, two of each outlier animals were removed for the analysis. So, we used 635 birds for GWAS of IMP and Inosine and 637 birds for GWAS of Hypoxanthine.

We added the information in the revised manuscript. In line 113 of the revised manuscript, “Finally, trimmed phenotypic values were used in an association test.” was modified into “Finally, 635 IMP and inosine contents and 637 hypoxanthine contents were filtered and used in an association test.”

Comment 6. Lines 151-153: please delete.

Response: We agree with the reviewer’s comment. Because lines 146-148 are enough to explain the information in lines 151-153 of the manuscript. To avoid redundancy, we removed the sentence “The GO and KEGG databases were explored via STRING using the protein-protein interaction (PPI) network.” from line 154 of the revised manuscript.

To use the abbreviation, the word “PPI” was modified to “protein-protein interaction (PPI)” in line 229 of the revised manuscript.

Comment 7. Lines 165-166: how do you calculate heritability?

Response: Thanks for the reviewer’s comment about the heritability calculation method. We added a sentence “And using the genetic relationship metrics and the preprocessed phenotype, we calculated the heritability of each trait through REML analysis of GCTA software.” in line 127 of the revised manuscript.

Comment 8. Figure 1: Please correct the headings of the histograms (A)-(B)-(C).

Response: Thank you for the comment. The headings of the histograms were corrected from “QQ plot for” to “Histogram for” in Figure 1.

Comment 9. Line 177: why you don’t calculate also the Bonferroni adjusted p>0.01 and reported this threshold in Figure 2?

Response: Thank you for the valuable comment. We added Bonferroni adjusted p<0.01 threshold line in Figure 2 and modified a sentence from “Bonferroni-corrected 5% significance threshold.” to “Blue and red lines indicate the Bonferroni-corrected 1% and 5% significance threshold respectively.” in line 207 of the revised manuscript.

Comment 10. Figure 2: please write that the Y-axis of panels (A), (B), (C), are different.

Response: Thanks for the reviewer’s comment. I added a sentence “The y-axis scale of (A), (B), and (C) are different.” in line 207 of the revised manuscript.

Comment 11. Figure 2: Is rs316338889 located in an exon or in an intron (or in another noncoding region) of DUSP8 gene?

Response: As we mentioned in lines 139 and 191 of the manuscript, we used GRCg6a 106 version genome annotation for the analysis. In this genome annotation version, the ENSEMBL database show that rs316338889 is located at chromosome 5:14370082 which is intron region of DUSP8 gene.

Comment 12. Line 191: Please indicate with more details the database you used because in the current release of NCBI (https://www.ncbi.nlm.nih.gov/genome/gdv/browser/genome/?id=GCF_016699485.2) and ENSEMBL (https://www.ensembl.org/Gallus_gallus/Gene/Summary?db=core;g=ENSGALG00010024637;r=5:13758675-13792985;t=ENSGALT00010060146) databases the gene DUSP8 and the marker rs316338889 (ENSEMBL only) are located outside the region you considered in figure 2(D).

Response: Thanks for your comment and concern. As we explained above, GRCg6a 106 genome version was used for all research period. And below link (ENSEMBL) is the information we used.

https://apr2022.archive.ensembl.org/Gallus_gallus/Variation/Mappings?db=core;r=5:14369582-14370582;v=rs316338889;vdb=variation;vf=1547324

Reviewer will find that the SNP location is in the intron region of DUSP8 gene using the link I provided above.

Comment 13. Line 206: Have you also considered the noncoding genes in the significant genomic region?

Response: Thanks for the question regarding the significant genomic region. There are 13 of LncRNA and MiRNA but they were not considered in this research. Because the functions of those noncoding genes are not clear.

Comment 14. Lines 230-233: please delete.

Response: Thank you for the concern of the reviewer. We removed the type “(Figure 4)” from line 236 of the revised manuscript. However, KEGG terms retrieved from STRING database are essential to support the point of the discussion. Therefore, we didn’t delete the other sentence that was suggested by the reviewer.

Comment 15. Line 248: rs31633888 > rs316338889

Response: Thank you for your comment. In line 257 of the revised manuscript “rs31633888” was corrected to “rs316338889”.

Comment 16. From line 260: have you considered all the protein-coding genes located within the significant genomic region for the functional analysis? Have you also considered the closest to the most significant markers?

Response: We considered all the protein-coding genes located within the significant genomic region for the functional analysis. And the putative candidate genes were selected based on the gene function, not the distance from the top significance marker. However, the marker is included in DUSP8 and C5NT1AL genes and those genes’ functions are supporting the association between the gene and the trait.

# Additional revision.

We added a funding resource information sentence “and also supported by Chungnam National University, South Korea.” in line 370 of the revised manuscript.

Reviewer 2 Report

Minjun Kim et al. explored candidate genes regulating nucleotide-related compounds included in chicken breast meat using a genome-wide association study. They found six candidate genes related to the function that possibly affected the content of nucleotide-related compounds in the muscle. The findings may be useful for chicken breeders, enabling them to better understand the underlying biological processes and identify potential genes and markers that could be used in breeding programs. Generally, the manuscript was well written and interesting, and I only have several minor suggestions.

1.        As for keywords, please remove 1, 2, 3, 4, and 5 after the names of keyword.

2.        Line 128, “g- is the effect of all 128 SNPs that were calculated using the genetic relationship metrics”, please revised g- according previous publication to indicate genetic relationship metrics, g- may not commonly used.

3.        Page 3, remove the sentences from line 151-153.

4.        Line 164, (p > 0.05)? what is means?have author performed test for normal distribution, which method? And what the statistic approach?

5.        3.2. GWAS results and the related genomic region > 3.2. GWAS analyses and the candidate genomic region.

6.        As for figure 1, please enlarge the size of x-axis and y-axis. The same for figure 2D.

7.        Remove line 230—233, “Authors should discuss the results and how they can be interpreted from the per…may also be highlighted.”

8.        Line 260 4.2. Functional analysis > 4.2. GO analysis, the same for 4.3 and 4.4.

9.        The authors have detected a list of candidate genes for the content of nucleotide- 2

related compounds in Korean native chicken, however, is there any other publication have been reported that these genes have been related to meat quality in other chicken breed, this may be included in the discussion.

10.      The 5. Conclusions can be shorted to highlight the most important points.

  The manuscript was well written. Minor editing of English language required

Author Response

[ Response to the reviewer 2’s comments addressed by point-by-point ]

 * Please see the attached revised manuscript.

General comment: Minjun Kim et al. explored candidate genes regulating nucleotide-related compounds included in chicken breast meat using a genome-wide association study. They found six candidate genes related to the function that possibly affected the content of nucleotide-related compounds in the muscle. The findings may be useful for chicken breeders, enabling them to better understand the underlying biological processes and identify potential genes and markers that could be used in breeding programs. Generally, the manuscript was well written and interesting, and I only have several minor suggestions.

Comment 1. As for keywords, please remove 1, 2, 3, 4, and 5 after the names of keyword.

Response: Thanks for the reviewer’s comment regarding keyword format. We removed the numbers after the names of keyword at lines 40-41 in the revised manuscript.

Comment 2. Line 128, “g- is the effect of all SNPs that were calculated using the genetic relationship metrics”, please revised g- according previous publication to indicate genetic relationship metrics, g- may not commonly used.

Response: Thanks for the detailed comment on the model. We agree that g- is not commonly used to indicate the genetic relationship metrics. However, we used the expression according to the manual of the software we applied (GCTA, https://yanglab.westlake.edu.cn/software/gcta/#MLMA). We used the command “--mlma-loco” for GWAS analysis and the corresponding linear equation is “y = a + bx + g- + e” as the equation of the manuscript. Authors think that using the same expression of the software manual is the most correct explanation. Therefore, we leave as is.

Comment 3.  Page 3, remove the sentences from line 151-153

Response: Thanks for the reviewer’s comment. Because lines 150-152 are enough to explain the information in lines 152-153 of the manuscript. To avoid redundancy, we removed the sentence “The GO and KEGG databases were explored via STRING using the protein-protein interaction (PPI) network.” from line 155 of the revised manuscript.

To use the abbreviation, the word “PPI” was modified to “protein-protein interaction (PPI)” in line 230 of the revised manuscript.

Comment 4. Line 164,(p > 0.05)? what is means?have author performed test for normal distribution, which method? And what the statistic approach?

Response: Thanks for the reviewer’s concern. In lines 110-112 of the manuscript, the sentence “The normalizing method that produced the highest Shapiro-Wilk p-value for normalized data among the methods was chosen” explains that the Shapiro-Wilk test was applied to check the normality and “p>0.05” indicate that the distribution of the phenotype follows the normal distribution.

Comment 5. 3.2. GWAS results and the related genomic region > 3.2. GWAS analyses and the candidate genomic region.

Response: Thanks for the reviewer’s suggestion. We agreed to the idea of the reviewer. We modified the subheading from “3.2. GWAS results and the related genomic region” to “3.2. GWAS results and the candidate genomic region” in line 185 of the revised manuscript.

Comment 6. As for figure 1, please enlarge the size of x-axis and y-axis. The same for figure 2D.

Response: Thanks for the comment. We modified figure 1 and figure 2D to enlarge the axis size.

Comment 7. Remove line 230—233, “Authors should discuss the results and how they can be interpreted from the per…may also be highlighted.”

Response: Thank you for the reviewer’s comment. The pointed sentences were removed in line 244 of the revised manuscript.

Comment 8. Line 260 4.2. Functional analysis > 4.2. GO analysis, the same for 4.3 and 4.4.

Response: Thanks for the valuable suggestion about the subheadings in the discussion part. We carefully say that not all those 4.2, 4.3, and 4.4 paragraphs are dealing with GO analysis results only. As we mentioned in lines 145-149, literature investigation and the protein interaction database search process were included to the functional analysis.

 The paragraph 4.2 explained the role of TNNT3 and TNNI2 genes supported by the GO analysis result and the related reference. On the other hand, 4.3 explained the role of DUSP8 gene with the location of the most significant marker and the reference paper related to the gene function. And 4.4 used KEGG and PPI database search results to support the discussion point about the role of C5NT1AL gene. Therefore, we think that the subheadings do not need to be modified and leave as is.

Comment 9. The authors have detected a list of candidate genes for the content of nucleotide- 2 related compounds in Korean native chicken, however, is there any other publication have been reported that these genes have been related to meat quality in other chicken breed, this may be included in the discussion.

Response: Thank you for the comment. If we show the other association study to support the function of candidate genes, it would be ideal. However, it was difficult to find genes related with the same trait of present research. Because the meat flavor based on the meat metabolites (nucleotide and amino acids etc.) is a novel trait in the field of chicken genetics. There is a recent *publication about chicken meat flavor using transcriptome analysis. But candidate genes reported in the paper and the present research are different because experimental conditions are not the same. Therefore, we didn’t put those supporting information in the discussion part.

* Identification of Key Genes Affecting Flavor Formation in Beijing-You Chicken Meat by Transcriptome and Metabolome Analyses (doi: https://doi.org/10.3390/foods12051025)

Comment 10. The 5. Conclusions can be shorted to highlight the most important points.

Response: Thank you for the valuable suggestion. We understand the point of the reviewer. However, we think that a brief summary of the findings of this study should be included. To shorten the words, the sentence “We conducted a GWAS to identify potential improvements in meat quality that could be achieved via KNC breeding. We conducted a GWAS and detected a genomic region on chromosome 5 that had a significant effect on the contents of nucleotide-related com-pounds in breast meat.” was modified to “We conducted a GWAS and detected a genomic region on chromosome 5 that had a significant effect on the contents of nucleotide-related compounds in breast meat.” in line 348 of the revised manuscript.

# Additional revision.

We added a funding resource information sentence “and also supported by Chungnam National University, South Korea.” in line 370 of the revised manuscript.

Round 2

Reviewer 1 Report

Please connect the sentences of lines 48-51 (meat quality) with lines 62-... (taste and flavor).

Please use the final line numbering in your comment.

Comment 12: what about the overlapping gene/transcript ENSGALG00000006608? Why don't you update (or consider) the current version of the chicken genome? If you still use version 106 please add the link you provided in the answer.

Author Response

[ Response to the reviewer 1’s minor revision comments ]

Comment 11. Please connect the sentences of lines 48-51 (meat quality) with lines 62-... (taste and flavor).

Response : We appreciate the reviewer providing a valuable comment enhancing the readability. To connect the sentences of lines the reviewer pointed, we moved the second paragraph of the introduction part to the lines 69-77 of revised manuscript.

Additional comment. Please use the final line numbering in your comment.

Response : Sure. In this comment, we are using line numbers of the revised manuscript of current version.

Comment 12: what about the overlapping gene/transcript ENSGALG00000006608? Why don't you update (or consider) the current version of the chicken genome? If you still use version 106 please add the link you provided in the answer.

Response : Thanks for the comment on the gene annotation of this research.

  • At first, gene ENSGALG00000006608 (ver 106; GRCg6a) the reviewer pointed is the same gene with ENSGALG00010024619 (ver 110; GRCg7b) which have gene name LOC770392 (as known as C5NT1AL) in current NCBI database. We confirmed the sequence similarity between gene ENSGALG00000006608 and ENSGALG00010024619 using BLAST tool of NCBI (Please see the result summary of the BLAST below and the attachment). And those two genes have almost the same adjacent genes in the genome region (Go to the link we provided below to see the genomic region of genes). Therefore, we used the gene name C5NT1AL instead of ENSGALG00000006608.

<Result summary of BLAST showing alignment of two genes sequences; ENSGALG00000006608 and ENSGALG00010024619>

Max Score

Total Score

Query Cover

E value

Per. Ident

Acc. Len

12480

12480

100%

0

99.19%

6920

Range 1: 1 to 6920

Identities:6881/6937(99%), Gaps:22/6937(0%), Strand: Plus/Plus

<Genomic region surrounding gene>

ENSGALG00000006608: https://asia.ensembl.org/Gallus_gallus_GCA_000002315.5/Location/View?db=core;g=ENSGALG00000006608;r=5:14310081-14315812;t=ENSGALT00000010667

ENSGALG00010024619:

http://asia.ensembl.org/Gallus_gallus/Location/View?db=core;g=ENSGALG00010024619;r=5:13720015-13725734;t=ENSGALT00010060075

  • Current study was conducted using the genome assembly of GRCg6a annotated with 106 genome version. Because Illumina chicken 60K SNP array which we used for the genotyping, provided the physical position of markers based on GRCg6a genome assembly version. Therefore, the following analyses were carried out in the same version of genome. We agree that if genes were annotated with the latest genome version, it would be effective to search for annotated genes. But we carefully think that GRCg6a version is not so outdated because most of the current GWAS research are using the GRCg6a reference genome, not the GRCg7b. Therefore, the genome version used in the current study is acceptable enough.
  • Here is the link to the Ensembl database of chicken reference assembly of GRCg6a annotated with 106 genome version.

https://apr2022.archive.ensembl.org/Gallus_gallus/Info/Index

 * Additional minor revisions are marked with red color in the revised manuscript.
